

# A molecular phylogeny of the Chinese *Sinopoda* spiders (Sparassidae, Heteropodinae): implications for taxonomy

He Zhang[1,*], Yang Zhong[2,*], Yang Zhu[1], Ingi Agnarsson[1,3] and Jie Liu[1,2]

[1] The State Key Laboratory of Biocatalysis and Enzyme Engineering of China, School of Life Sciences, Hubei University, Wuhan, Hubei, China
[2] School of Nuclear Technology and Chemistry & Biology, Hubei University of Science and Technology, Xianning, Hubei, China
[3] Department of Biology, University of Vermont, Burlington, VT, United States of America
[*] These authors contributed equally to this work.

## ABSTRACT

*Sinopoda* spiders are a diverse group with limited dispersal ability. They are remarkably sympatric among related species, which often results in misidentification and incorrect matching of sexes. In order to understand the evolutionary relationships and revise the taxonomy problems in this genus, we offer the first molecular phylogeny of *Sinopoda*. Our results strongly support the monophyly of *Sinopoda* and its sister relationship with *Spariolenus* and reject the monophyly of the *S. okinawana* species group. We establish three new species groups based on both molecular and morphological data. Our phylogeny also illuminates some taxonomic issues and clarifies some species limits: (1) Supporting the newly revised matching of sexes in *S. longiducta* and *S. yaanensis* by *Zhong et al. (2019)*. (2) The original description of *S. campanacea* was based on mismatched sexes. *S. changde* is proposed as a junior synonymy of *S. campanacea*, while the original female '*S. campanacea*' is here described as a new species: *S. papilionaceous* Liu **sp. nov.** (3) The type series of *S. serpentembolus* contains mismatched sexes. The female is considered as *S. campanacea*, while we here report the correctly matched females of *S. serpentembolus*. (4) We describe one additional new species: *S. wuyiensis* Liu **sp. nov.** Our first molecular phylogeny of *Sinopoda* provides a tool for comparative analyses and a solid base for the future biodiversity and taxonomic work on the genus.

# INTRODUCTION

Taxonomy can be challenging in groups that have undergone relatively little morphological change through the speciation processes. This is especially true when there is extensive sympatry among related species, a rather rare phenomenon (*Agnarsson et al., 2016*). While the majority of spider taxonomy is still purely based on morphological data, integrative approaches are critical to address taxonomy in such challenging groups (*Godwin & Bond,*

Corresponding authors
Ingi Agnarsson,
iagnarsson@gmail.com
Jie Liu, sparassidae@aliyun.com

*2021*). A good example of the kind of taxonomic problems readily solved by molecular phylogenetics is the correct matching of sexes in similar sympatric species.

With 133 species, *Sinopoda* Jäger, 1999, is the fourth largest genus of the family Sparassidae. China is the hotspot of diversity and endemicity of this genus, it is also widely distributed in East Asia (71 species in China, 12 species in Japan and Korea), Southeast Asia (50 species in Brunei, Indonesia, Laos, Malaysia, Myanmar, Thailand and Vietnam) and South Asia (one species in India). *Sinopoda* spiders prefer humid habitats in mountainous forests, and are common in leaf litter, rock crevices, caves, and on tree bark (*Grall & Jäger, 2020*; *Jäger, 1999*; *Jäger, 2012*; *Liu, Li & Jäger, 2008*; *Zhang, Zhang & Zhang, 2015*). Some cave dwelling *Sinopoda* spiders (such as *S. guap* Jäger, 2012, *S. steineri* Jäger, 2012 and others) exhibit obviously troglomorphic features including reduced eyes and pale body. Among them, *S. caeca* Grall & Jäger, 2020 and *S. scurion* Jäger, 2012 are the only known blind (eyeless) huntsman spiders in the world. With preference for cryptic habitats *Sinopoda* spiders do not appear to engage in frequent long distance dispersal and have not been directly recorded dispersing through air on silk threads or ballooning (*Bell et al., 2005*).

*Sinopoda* spiders, as nocturnal hunters, are often known only by single sex due to the difficulty to collect mature pairs in the field. To date, over half of the known species (66) were described based on a single sex (*World Spider Catalog, 2021*), and the matching sex remains unknown. In addition, sympatry among species is extensive, and multiple species can be collected at a single site such as Hengduan Mountains and mountains surrounding the Sichuan Basin (Fig. 1). Therefore, mismatches or wrong identifications are common in this genus. For example, *Zhong et al. (2019)* considered *S. longiducta* as mismatched and transferred the female to *S. yaanensis* based on collection data and morphological characters. However, neither match has ever been tested phylogenetically.

To date, no molecular phylogeny has been published for this genus, or any group within it. A single study (*Moradmand, Schönhofer & Jäger, 2014*) included one *Sinopoda* species grouping it with the genera *Heteropoda* Latreill, 1804 and *Spariolenus* Simon, 1880. The monophyly of this genus and its known species groups (*S. chiangmaiensis*-group and *S. okinawana*-group) (*Grall & Jäger, 2020*; *Jäger & Ono, 2002*) remains to be untested. *Sinopoda* (the prefix "*sino*" means "belonging to China"), unsurprisingly has its center of diversity in China. In the past ten years, a series of surveys on Chinese *Sinopoda* spiders were conducted by the team from Hubei University and yielded numerous specimens of known and new species. This is our fourth paper on Chinese *Sinopoda* spiders (*Zhong, Cao & Liu, 2017*; *Zhong et al., 2018*; *Zhong et al., 2019*) and provides the first molecular phylogenetic estimate of *Sinopoda* spiders, with the following aims: (1) To test the monophyly of *Sinopoda* and *S. okinawana*-group respectively. (2) To investigate phylogenetic relationships among species and establish new species groups by combining molecular and morphological evidences. (3) To revise matching of sexes in putatively mismatched species and report on potentially new species based on molecular and morphological evidence.

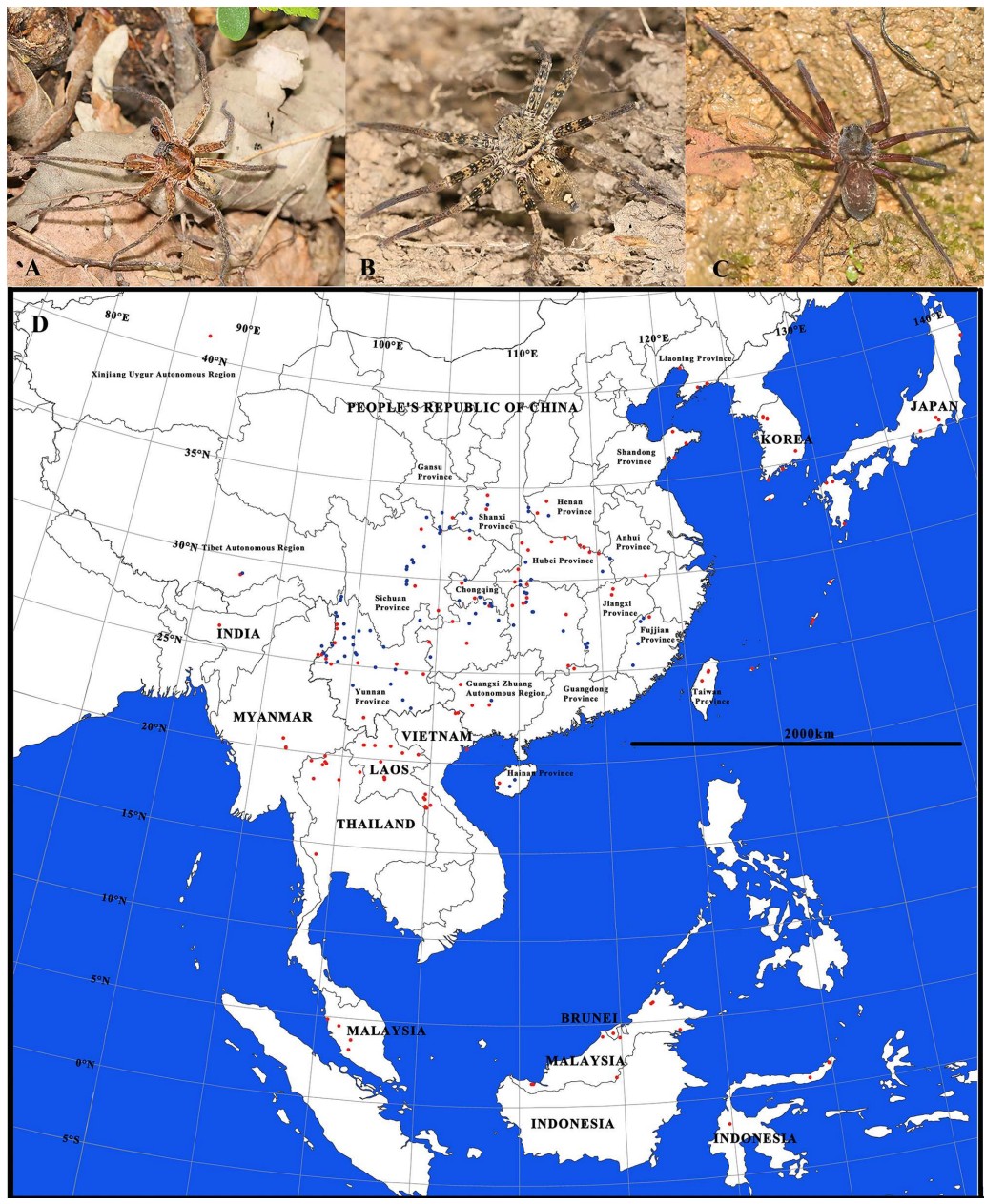

**Figure 1** **Habitus of *Sinopoda* spiders (A–C) and a map of localities where the known species of *Sinopoda* distributed in the world (D).** Every dot represents one locality. Red color represents the localities did not include in our analyses. Blue color represents the sample collection place involved in this project.

## MATERIALS & METHODS

### Taxon sampling

Spiders were sampled from China between 2008 and 2018 and deposited in the Centre for Behavioural Ecology and Evolution (CBEE), College of Life Sciences, Hubei University.

Most of these individuals were collected by the members of our laboratory and others were provided by the colleagues from Southwest University. A total of 856 individuals from 12 Provinces (Fujian, Gansu, Guizhou, Hainan, Henan, Hubei, Hunan, Jiangxi, Liaoning, Shanxi, Sichuan and Yunnan), one Municipality (Chongqing) and 1 Autonomous Regions (Xizang Autonomous Region) were collected from the field. Every specimen was given a unique identification number ('S' number). Species were initially sorted by morphological characteristics and stored in 70% ethanol for morphological work and in 100% ethanol for molecular analyses. In total, we included 70 specimens's sequences of the genus *Sinopoda* to molecular analyses including three individuals from Genbank. Individual data (including species name, sample locations and GenBank Accession Numbers) are provided in Supplement 1.

In the present paper, 10 species from 13 individuals were used as outgroups (including *Barylestis occidentalis* (Simon 1887), *H. davidbowie* Jäger, 2008, *H. jugulans* (L. Koch, 1876), *H. languida* Simon, 1887, *H. renibulbis* Davies, 1994, *H. venatoria* (Linnaeus, 1767), *Pandercetes* sp., *Pseudopoda confusa* Jäger et al., 2006, *P. prompta* (O. Pickard-Cambridge, 1885), *Spariolenus iranomaximus* Moradmand, & Jäger, 2011). This choice was guided by the recent phylogenetic result (*Moradmand, Schönhofer & Jäger, 2014*) and these five genera all belong to the subfamily Heteropodinae. We retrieved molecular data on 16 species from Genbank (Supplement 1).

## Molecular protocols

One or two legs of each individual (depending on the size of specimens) were used to extract total genomic DNA. DNA extraction was achieved with the Universal Genomic DNA Kit (CWBIO, Beijing, China). We used a target gene approach including both mitochondrial and nuclear genes. Six loci were targeted with different degrees of variability. Two mitochondrial genes (two regions including 16S ribosomal RNA gene (16S) and cytochrome c oxidase subunit 1 (COI)) and four nuclear genes (protein-coding histone H3 (H3), 18S ribosomal RNA gene (18S), 28S ribosomal RNA gene (28S) and Internal Transcribed Spacer 2 (ITS2)) were used in this research. Primers (*Folmer et al., 1994*; *Simon et al., 1994*; *White et al., 1990*) and PCR conditions are shown in Table 1. Multiple primers were employed in the amplification of a large region of COI (approximately 1.2 kb). These primers include the pairs LCOI1490 and HCOI2198, and Jerry and C1-N-2776. Fragments were sequenced by the companies of Tsingke Biological Technology (Wuhan, China) and Sunny Biotechnology Company Limited (Shanghai, China) in both directions. Sequences were assembled and edited using the Chromaseq module in Mesquite (*Maddison & Maddison, 2011a*; *Maddison & Maddison, 2011b*) employing Phred and Phrap (*Green, 1999*; *Green & Ewing, 2002*). After assembly, to all sequences were blasted against Genbank (National Center for Biotechnology Information (NCBI)) to verify they all belonged to the family Sparassidae.

## Phylogenetic analyses

All sequences were aligned with MAFFT (*Katoh, 2013*) on XSEDE in parallel on the Cyberinfrastructure for Phylogenetic Research Project (CIPRES Science Gateway) at the

Zhang et al. (2021), *PeerJ*, DOI 10.7717/peerj.11775
**Table 1   Molecular markers and primers used for amplification.** The amplification was performed in 50 $\mu$l final volume containing 18 $\mu$l of ultra-pure water (dd H$_2$O), 25 $\mu$l of I-5$^{TM}$ 2X High-Fidelity Master Mix, 2 $\mu$l of each primer (100 pmol/ $\mu$l), 3 $\mu$l of the genomic spider DNA templates. PCR settings list Initial Denaturation (ind), followed by /n cycles (Denaturation: de, Primer Annealing: pra, Primer Elongation: pre), and one Terminal Elongation (tee). (Temperature in °C following by time in seconds).

| Marker | Primer name | Premier sequence (5′→ 3′) | PCR settings |
|---|---|---|---|
| 16S | 16SA | CGCCTGTTTACCAAAAACAT | ind98 (120s), [de 98(10s), pra 52(15s), pre72(15s)/35], tee72 (120s) |
| | 16SB | CCGGTTTGAACTCAGATC | |
| 18S | 18S5f | GCGAAAGCATTTGCCAAGAA | ind98 (120s), [de 98(10s), pra 57(15s), pre72(15s)/35], tee72 (120s) |
| | 18S9r | GATCCTTCCGCAGGTTCACCTAC | |
| 28S | 28SC | GGTTCGATTAGTCTTTCGCC | ind98 (120s), [de 98(10s), pra 55(15s), pre72(15s)/35], tee72 (120s) |
| | 28SO | GAAACTGCTCAAAGGTAAACGG | |
| COI | LCOI1490 | GGTCAACAAATCATAAAGATATTGG | ind98 (120s), [de 98(10s), pra 47(15s), pre72(15s)/35], tee72 (120s) |
| | HCOI2198 | TAAACTTCAGGGTGACCAAAAAATCA | |
| | Jerry | CAACATTTATTTTGATTTTTTGG | ind98 (120s), [de 98(10s), pra 52(15s), pre72(15s)/35], tee72 (120s) |
| | C1-N-2776 | GGATAATCAGAATATCGTCGAGG | |
| H3 | H3aF | ATGGCTCGTACCAAGCAGACVGC | ind98 (120s), [de 98(10s), pra 56(15s), pre72(15s)/35], tee72 (120s) |
| | H3aR | ATATCCTTRGGCATRATRGTGAC | |
| ITS2 | ITS4 | TCCTCCGCTTATTGATATGC | ind98 (120s), [de 98(10s), pra 50(15s), pre72(15s)/35], tee72 (120s) |
| | ITS5.8 | GGGACGATGAAGAACGCAGC | |

UC San Diego Supercomputing Center (*Miller, Pfeiffer & Schwartz, 2010*). Other large analyses were performed also using this platform.

Considering the lack of gaps, we used the L-INS-i method to align the protein-coding genes H3 and COI. We verified absence of stop codons by translating sequences to amino acids. In virtue of the highly variable structure of ribosomal RNA genes, the ambiguously aligned regions were excluded by using the E-INS-i method to align the following four genes: 16S, 18S, 28S, ITS2 (*Wheeler et al., 2016*). We concatenated these six aligned genes in Mesquite.

Two analytical methods (Maximun Likelihood and Bayesian) were used to estimate the phylogenetic relationships. In all analyses, we treated the gaps and ambiguous as missing data. Trees for all target genes were also reconstructed. Bayesian inference analyses were performed via the parallel MrBayes 3.2.6 (*Ronquist et al., 2012*) on XSEDE. Due to the highly substitution rates of the third position, protein-coding genes (COI and H3) were implemented three different partition schemes, namely as COI-1st, COI-2nd, COI-3rd, H3-1st, H3-2nd and H3-3rd. For sensitivity analyses of the multilocus dataset, six genes were divided into ten data partitions, the jModelTest2 on XSEDE (2.1.6) (*Darriba et al., 2012*) were used to choose the most suitable and best-fit models for mtDNA and nuDNA, according to the Akaike information criterion (AIC) (*Posada & Buckley, 2004*). The model parameters were estimated during the analyses and the choice by the jModelTest2 on XSEDE (2.1.6). For 16S, 28S, COI-2nd and ITS2, we used the model of GTR + I + G. The best model for 18S is GTR. GTR + G for COI-3rd and H3-1st. HKY + I + G model were used for the partitions of COI-1st. HKY + G for H3-3rd. For H3-2nd, we used the model of HKY. For every analysis, $5{\star}10^7$ generations were run for two simultaneous independent analyses with four Markov Chains (one cold and three heated) and every 1000th states were saved for the current tree file. Based on the TRACER v1.7.1 (*Rambaut & Drummond, 2007*), all the results for the posterior distributions of the parameters had an Effective Sample Size (ESS) $\geq$ 200. The first 25% trees ($1.25 \times 10^7$ generations) of every run were discarded as burn in. Maximum likelihood (ML) analyses were performed using IQ-Tree on XSEDE (*Nguyen et al., 2015*) on the focal matrix with same partitions as implemented in the Bayesian analysis. Node support was estimated using ultrafast bootstrapping with 1,000 replicates (*Hoang et al., 2018*) and the Shimodaira-Hase-gawa approximate likelihood-ratio test (SH-aLRT) with 1,000 replicates (*Guindon et al., 2011*).

### Taxonomy

Specimens were examined with an Olympus SZX16 stereomicroscope; details were further investigated with an Olympus BX51 compound microscope. Epigyne were cleared in proteinase K at 56 °C to dissolve non-chitinous tissues. Photos were taken with Leica M205C stereomicroscope and Olympus BX51 equipped with a Micropublisher 3.3 RTV camera (QImaging, Surrey, BC, Canada). The digital images depicting the habitus and genital morphology were a composite of multiple images taken at different focal planes along the *Z* axis and assembled using the software package Helicon Focus 3.10.

Leg measurements are shown as: total length (femur, patella, tibia, metatarsus, tarsus). Numbers of spines are listed for each segment in the following order: prolateral, dorsal,
retrolateral, ventral (in femora and patellae ventral spines are absent and fourth digit is omitted in the spination formula). All measurements are in millimeters.

## Nomenclatural acts

According to the International Commission on Zoological Nomenclature (ICZN), the electronic version of this article in portable document format (PDF) will represent a published work. The new species names contained in the electronic version are effectively published under that Code from the electronic edition alone. This article and the nomenclatural acts it contains have been registered in ZooBank, the online registration system for the ICZN. The ZooBank LSIDs (Life Science Identifiers) can be resolved and the associated information viewed through any standard web browser by appending the LSID to the prefix http://zoobank.org/. The LSID for this publication is: urn:lsid:zoobank.org:pub: DE32C06B-FB70-497D-A690-74DED52939DB.

## PHYLOGENETIC RESULTS

Our full DNA matrix contains 83 individuals, 70 of which belongs to 38 *Sinopoda* species including about one-third (29%) of the known species and one new species. For outgroups we included 13 individuals belonging to 10 species in five genera of the subfamily Heteropodinae. The aligned sequences amounted to 460 bp for 16S (65 individuals), 818bp for 18S (42 individuals), 698bp for 28S (79 individuals), 1155bp for COI (79 individuals), 330bp for H3 (74 individuals), 386bp for ITS2 (66 individuals). The phylogenetic trees from the two phylogenetic methods (Bayesian inference and Maximum Likelihood) were highly consistent with relatively high posterior probabilities (PP) and bootstrap values (BS). Hence, we showed the nodal supports with these two analyses together (Fig. 2) on the BI topology. In general, Bayesian posterior probabilities were slightly higher than ML bootstrap supports. The monophyly of *Sinopoda* was robustly supported (PP 1.00, BS 100%). *Spariolenus* was supported as the sister group of *Sinopoda* (PP 0.93, BS 64%), which is not consistent with former study (*Moradmand, Schönhofer & Jäger, 2014*). The phylogeny suggested the polyphyly of the *S. okinawana*-group, as *S. nuda* is far removed from the remaining group members. We established three new species groups, according to the phylogeny, all supported by morphological and molecular characters: *S. anguina*-group (Fig. 3, PP 1.00, BS 97%), *S. globosa*-group (Fig. 4, PP 0.97, BS lower than 50%) and *S. tumefacta*-group (Fig. 5, PP 1.00, BS 100%). These are diagnosed, described and illustrated in detail in the following taxonomy part. Male and female individuals from the Tianping Mountain belonging to *S. campanacea* and *S. changde* were analysed. Male *S. campanacea* form a monophylum with the female *S. changde* (PP 0.98, BS 94%). The originally described female of *S. campanacea* is considered as a new species: *S. papilionaceous* Liu **sp. nov.** which is redescribed in current paper according to the original illustration (*Wang, 1990*: 7, Figs. 4–5). For *S. serpentembolus*, the male of *S. serpentembolus* from Shanxi Province and female individuals from Henan Province are monophyletic (PP 1.00, BS 100%). However, females differ significantly from the originally matched female of *S. serpentembolus* (*Zhang et al., 2007*: 251, Figs. 5–6) based on morphological data. Meanwhile, our result support the recent revised matching of sexes in *S. longiducta* and *S. yaanensis* by *Zhong et al. (2019)* as

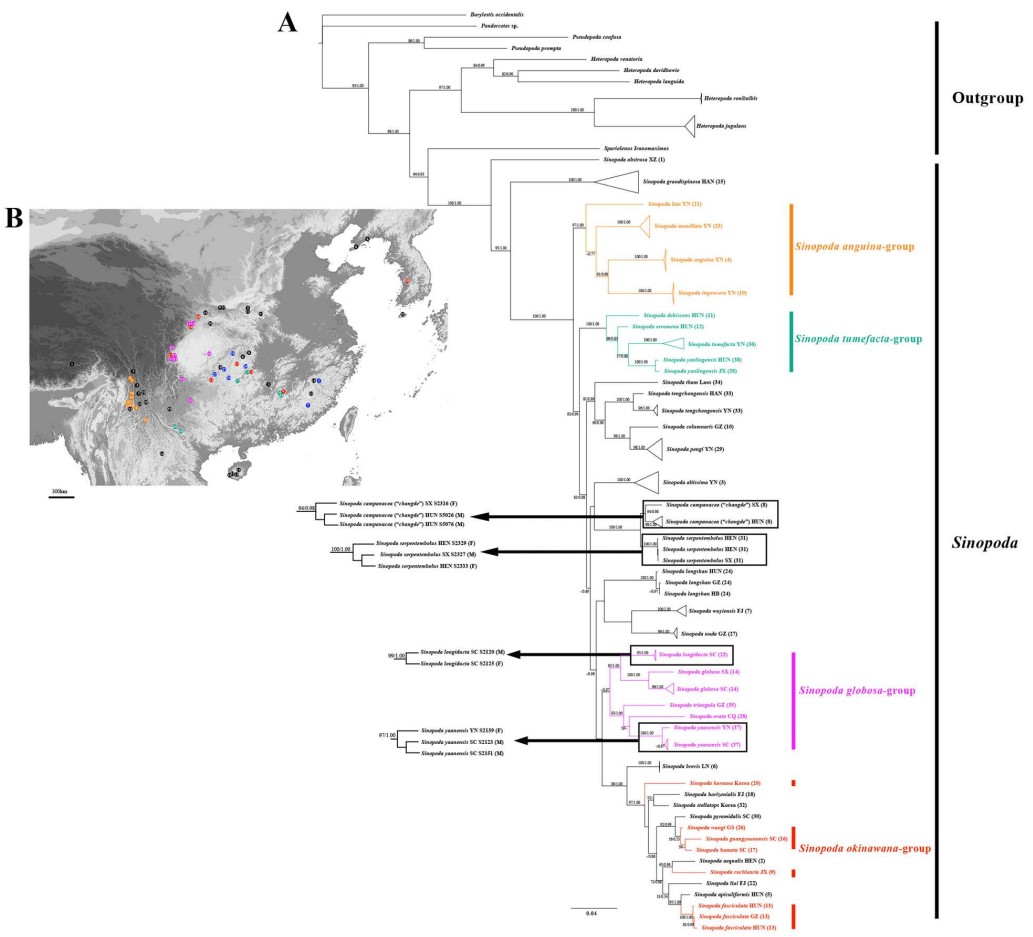

**Figure 2  Summary of phylogenetic analysis of the genus *Sinopoda* and the sample current distribution in Asia.** (A) Combined results of the phylogenetic analysis based on six gene fragments. The numbers at the nodes represent bootstrap support values/posterior probabilities from both likelihood and Bayesian analyses. (B) Map with sampling localities of the genus *Sinopoda* and its major lineages. Different colors refer to different groups. *S. anguina*-group (orange), *S. globosa*-group (purple), *S. tumefacta*-group (cyan) and *S. okinawana*-group (red). (1. *S. abstrusa* Zhong et al., 2019; 2. *S. aequalis* Zhong et al., 2019; 3. *S. altissima* (Hu & Li 1987); 4. *S. anguina*; 5. *S. apiculiformis* Zhong et al., 2019; 6. *S. brevis* Zhong et al., 2019; 7. *S. wuyiensis* Agnarsson & Liu sp. nov.; 8. *S. campanacea* ("*changde*"); 9. *S. cochlearia* Zhang, Zhang & Zhang, 2015; 10. *S. columnaris* Zhong et al., 2019; 11. *S. dehiscens*; 12. *S. erromena*; 13. *S. fasciculata* (Jager, Gao & Fei, 2002); 14. *S. globosa*; 15. *S. grandispinosa* Liu, Li & Jäger, 2008; 16. *S. guangyuanensis* Zhong et al., 2018; 17. *S. hamata* (Fox, 1937); 18. *S. horizontalis* Zhong, Cao & Liu, 2017; 19. *S. improcera*; 20. *S. koreana* (Paik, 1968); 21. *S. lata*; 22. *S. liui* Zhong, Cao & Liu, 2017; 23. *S. longiducta*; 24. *S. longshan*; 25. *S. mamillata*; 26. *S. wangi* Song & Zhu, 1999; 27. *S. nuda*; 28. *S. ovata*; 29. *S. pengi* Song & Zhu, 1999; 30. *S. pyramidalis* Zhong et al., 2019; 31. *S. serpentembolus*; 32. *S. stellatops* Jäger & Ono, 2002; 33. *S. tengchongensis* Fu & Zhu, 2008; 34. *S. tham* Jäger, 2012; 35. *S. triangula*; 36. *S. tumefacta*; 37. *S. yaanensis*; 38. *S. yanlingensis*).

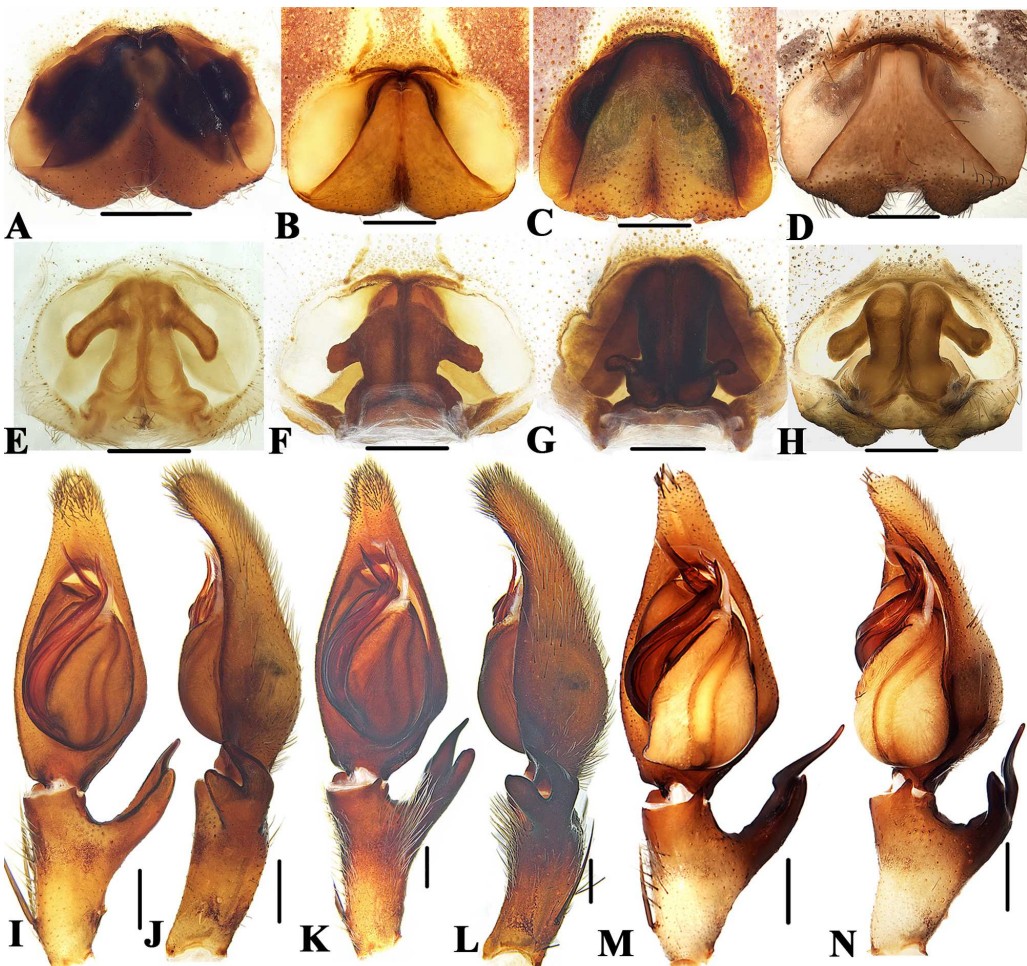

**Figure 3** **Members of *Sinopoda anguina*-group.** *Sinopoda anguina* ((A), female epigyne, ventral; (E), female vulva, dorsal; (I), left male palp, ventral; (J), left male palp, retrolateral); *Sinopoda improcea* ((B), female epigyne, ventral; (F), female vulva, dorsal; (K), left male palp, ventral; (L), left male palp, retrolateral); *Sinopoda lata* ((C), female epigyne, ventral; (G), female vulva, dorsal); *Sinopoda mamillata* ((D), female epigyne, ventral; (H), female vulva, dorsal; (M), left male palp, ventral; (N), left male palp, retrolateral). Scale bars: 0.5 mm.

both species are monophyletic (*S. longiducta* PP 1.00, BS 95%; *S. yaanensis* PP 1.00, BS 100%).

## DISCUSSION

We provide the first phylogenetic analysis of *Sinopoda*, focusing on the Chinese species. Our analysis strongly supports the monophyly of *Sinopoda*. According to a previous study (*Moradmand, Schönhofer & Jäger, 2014*), *Sinopoda* was hypothesized to group with *Heteropoda* and *Spariolenus*. We find support to the sister relationship between *Sinopoda* and *Spariolenus*, however, further sampling of Heteropodinae genera will be necessary to further clarify the placement of *Sinopoda*.

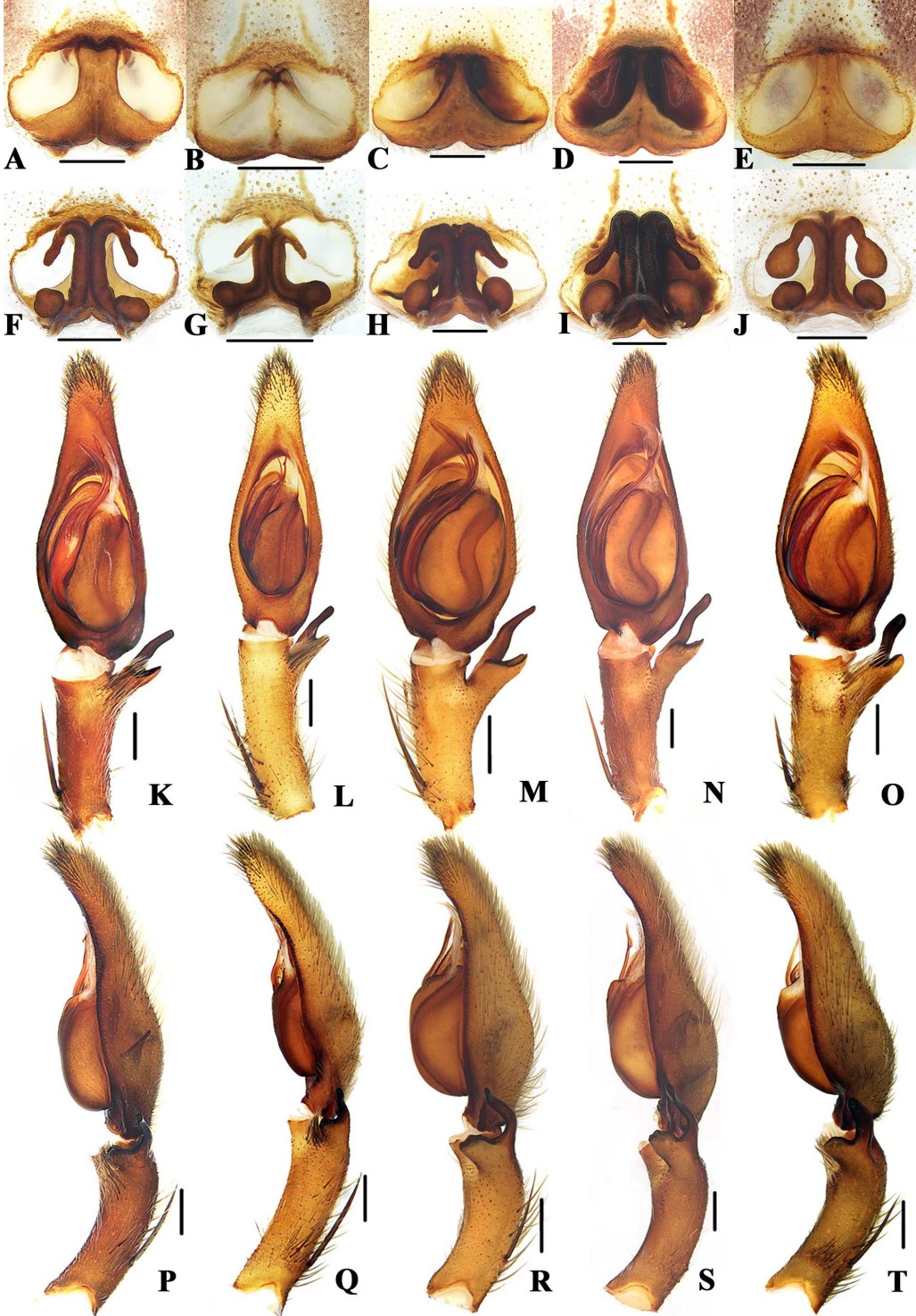

**Figure 4 Members of *Sinopoda globosa*-group.** *Sinopoda globosa* ((A), female epigyne, ventral; (F), female vulva, dorsal; (K), left male palp, ventral; (P), left male palp, retrolateral); *Sinopoda longiducta* ((B), female epigyne, ventral; (G), female vulva, dorsal; (L), left male palp, ventral; (Q), left male palp, retrolateral); *Sinopoda ovata* ((C), female epigyne, ventral; H, female vulva, dorsal; (M), left male palp, ventral; (R), left male palp, retrolateral); *Sinopoda triangula* ((D), female epigyne, ventral; (I), female vulva, dorsal; (N), left male palp, ventral; (S), left male palp, retrolateral); *Sinopoda yaanensis* ((E), female epigyne, ventral; (J), female vulva, dorsal; (I), left male palp, ventral; (J), left male palp, retrolateral). Scale bars: 0.5 mm.

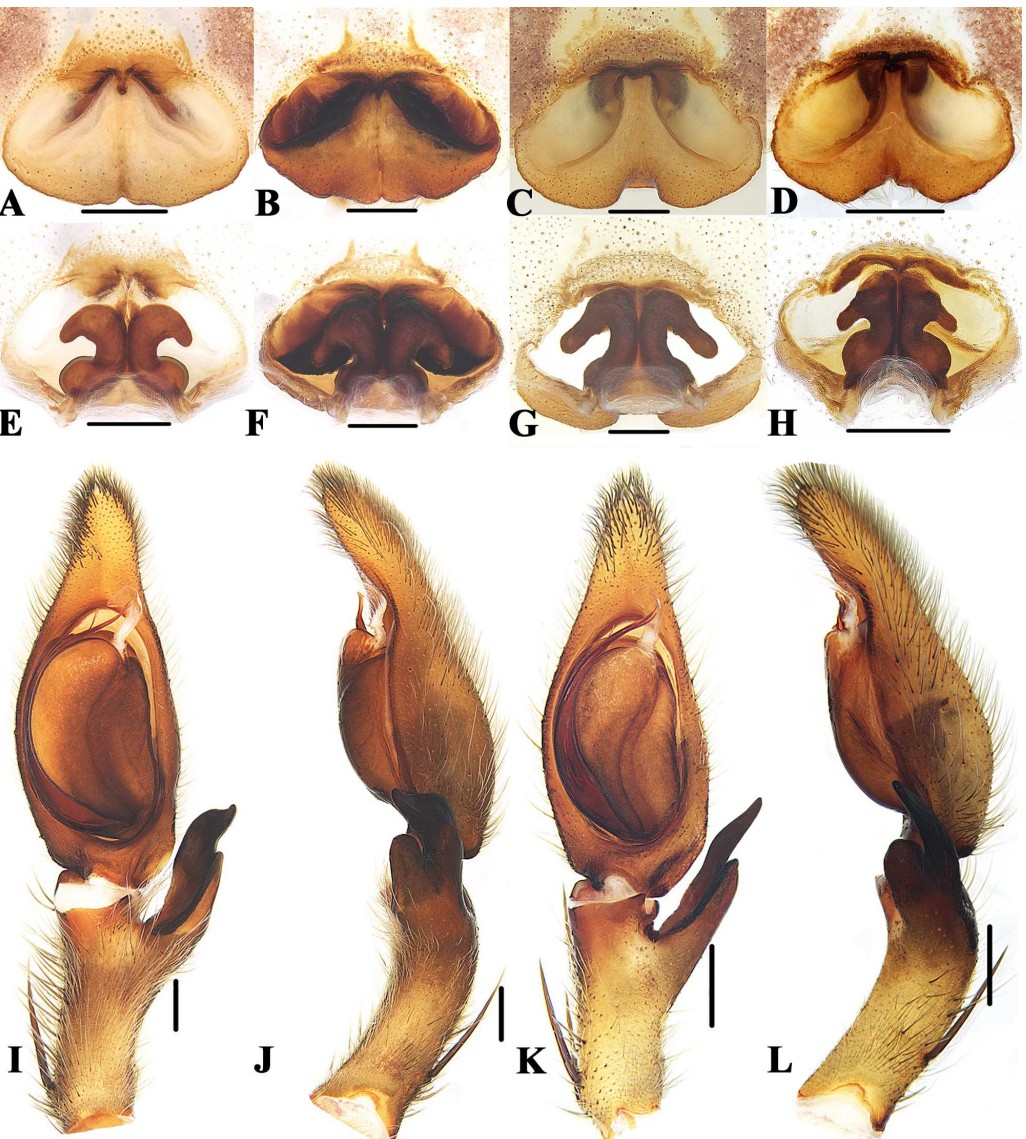

**Figure 5** **Members of *Sinopoda tumefacta*-group.** *Sinopoda dehiscens* ((A), female epigyne, ventral; (E), female vulva, dorsal); *Sinopoda erromera* ((B), female epigyne, ventral; (F), female vulva, dorsal); *Sinopoda tumefacta* ((C), female epigyne, ventral; (G), female vulva, dorsal; (I), left male palp, ventral; (J), left male palp, retrolateral); *Sinopoda yanlingensis* ((D), female epigyne, ventral; (H), female vulva, dorsal; (K), left male palp, ventral; (L), left male palp, retrolateral). Scale bars: 0.5 mm.

Since the taxonomy of *Sinopoda* is poorly known and hitherto little genetic data has been available, species identification and matching of sexes has been challenging and has relied chiefly on field data. However, because many species share very similar morphology and due to extensive species sympatry, misidentifications are common and mismatches of sexes are common in *Sinopoda*. The sex matching of *Sinopoda* spiders is guided by field data and morphology but should in general be tested using molecular evidence. Such molecular testing of hypothesis is absolutely critical in cases of highly similar sympatric species.

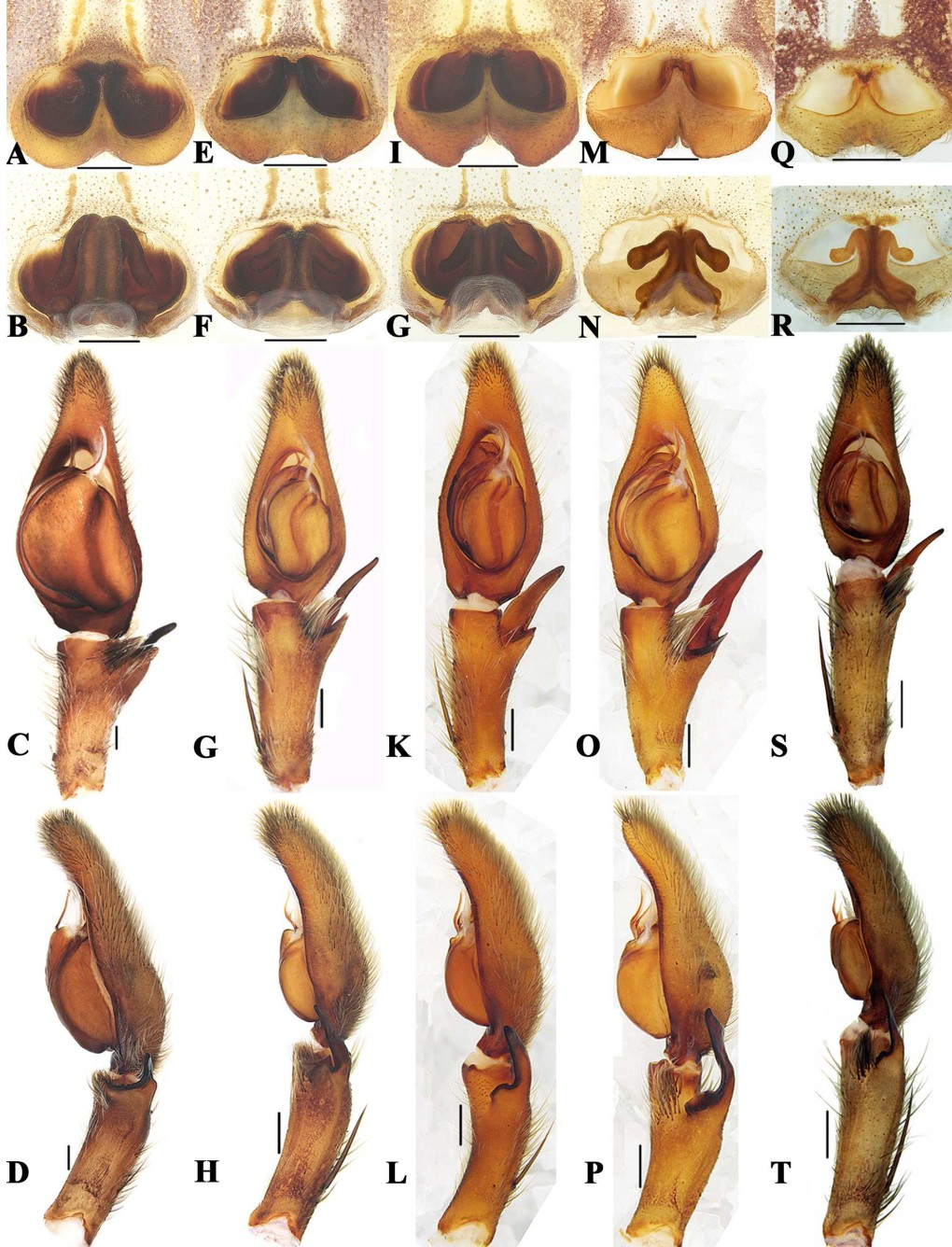

**Figure 6** **Members of *Sinopoda okinawana*-group (collected in this project).** *Sinopoda cochlearia* ((A), female epigyne, ventral; (B), female vulva, dorsal; (C), left male palp, ventral; (D), left male palp, retrolateral); *Sinopoda fasciculata* ((E), female epigyne, ventral; (F), female vulva, dorsal; (G), left male palp, ventral; (H), left male palp, retrolateral); *Sinopoda guangyuanensis* ((I), female epigyne, ventral; (J), female vulva, dorsal; (K), left male palp, ventral; (L), left male palp, retrolateral); *Sinopoda hamata* ((M), female epigyne, ventral; (N), female vulva, dorsal; (O), left male palp, ventral; (P), left male palp, retrolateral); *Sinopoda wangi* ((Q), female epigyne, ventral; R, female vulva, dorsal; S, left male palp, ventral; (T), left male palp, retrolateral). Scale bars: 0.5 mm.

Our phylogeny and DNA evidence clarify the taxonomy and species level classification of *Sinopoda* spiders, and tests and clarifies recent taxonomic rematching of sexes (*Zhang, Zhang & Zhang, 2015*; *Zhong et al., 2019*). Our study also provides strong evidence for further rematching of sexes and recircumscription of some species, and the results in the description of two new species. The *S. campanacea* holotype is male, we therefore propose *S. changde* as a **new synonym** of *S. campanacea*. The originally described female of *S. campanacea* is considered as a new species: *S. papilionaceous* Liu **sp. nov.** which is redescribed in current paper according to the original illustration (*Wang, 1990*: 7, Figs. 4–5). For *S. serpentembolus*, the male of *S. serpentembolus* from Shanxi Province and female individuals from Henan Province are monophyletic (PP 1.00, BS 100%). However, females differ significantly from the originally matched female of *S. serpentembolus* (*Zhang et al., 2007*: 251, Figs. 5–6) based on morphological data. Therefore, we revised the *S. serpentembolus* as following: (1) The correct female of *S. serpentembolus* is reported for the first time. (2) The originally mismatched female of *S. serpentembolus* is found to be similar to *S. campanacea* where we tentatively place it. We note, however, that it shows some morphological differences with *S. campanacea* and further molecular data is needed to clarify its placement.

In addition to providing crucial matching information on species our phylogeny aids in the organization of species into species groups that can be useful units for evolutionary and biogeographical questions. On the basis of morphological resemblance, many *Sinopoda* species have been classified into different groups. We established three new groups: *S. anguina*-group, *S. globosa*-group and *S. tumefacta*-group basing on both molecular and morphological data (Figs. 2–5). These groups have their unique biogeography, although some have partially overlapping geographical distribution. Members of *S. anguina*-group, which includes 12 species (only four are included in our analysis) distributed in southern area of the Ailao Shan-Red River Fault zone (Southeast Asia, Hengduan Moutains of Yunnan). Members of *S. globosa*-group are distributed in the mountains surrounding the Sichuan Basin.

Our analysis indicates that the morphologically-based "*S. okinawana*-group" (Fig. 6) is polyphyletic. Because diagnostic characters for the females are weak (*Jäger & Ono, 2002*; *Liu, Li & Jäger, 2008*; *Zhong et al., 2018*), this group was original diagnosed mainly based on male genital characters, especially the reduced embolic apophysis and ventral RTA. However, the reduced embolic apophysis is homoplastic and also occurs in other *Sinopoda* species such as *S. tumefacta* and *S. yaanensis*. In addition, the embolic apophysis is totally absent in *S. longshan* Yin et al., 2000 and *S. nuda* Liu, Li & Jäger, 2008. These evidences suggest that the embolic apophysis evolves rapidly enough in this genus to not be reliable for group diagnostics. Our observations of the RTA, and prior studies, suggest that it may also evolve rapidly. In general, the characters of male palp may not be the best evidence to guide classification at the higher level due to rapid evolution (*Eberhard, 2004*). This conclusion is consistent with the finding in Australian huntsman spiders (*Agnarsson & Rayor, 2013*). A further revision of the "*S. okinawana*-group" is necessary including more species and other putative synapomorphies. It seems likely that at least a portion of the current group

will form a clade as almost all members of "*S. okinawana*-group" are distributed in similar low-altitude areas, such as Northeast China Plain (*Zhang et al., submitted*).

In conclusion, we argue that integrative taxonomy—approaching species delimitation and description using multiple lines of evidence (*Godwin & Bond, 2021*)—is critical in *Sinopoda*, a diverse genus of morphologically similar spiders. Our study provides a solid base for understanding the biodiversity and taxonomy of *Sinopoda*, and a tool for comparative studies, such as analyses of biogeographical patterns in this genus. Nevertheless, we highlight that our only contains about a third of all the known species and that some of the deeper nodes, and several of the shallower clades in this phylogeny, are weakly supported. Further assessment of *Sinopoda* biodiversity and phylogeny further sampling of species, and updated genetic approaches through next generation sequencing techniques will be necessary.

## Taxonomy

**Family Sparassidae Bertkau, 1872**
**Subfamily Heteropodinae Thorell, 1873**

**Genus *Sinopoda* Jäger, 1999**
**Type species:** *Sarotes forcipatus* (Karsch, 1881)

**Diagnosis:** Small to large spiders with laterigrade legs. Male palp with bifurcated RTA (retrolateral tibial apophysis), membranous conductor which is arising from distal part of tegulum, with embolic apophysis in most species. Female epigynum with modified epigynal rims and distinct lobal septum. Female vulva with internal duct system fused along median line which is divided into a basal part and a head, situated laterally from the entrance of copulatory ducts into internal duct system (*Grall & Jäger, 2020*; *Jäger, 1999*; *Liu, Li & Jäger, 2008*; *Zhang, Zhang & Zhang, 2015*).

### *Sinopoda anguina*-group
(Figure 3)
**Diagnosis.** This group can be recognized by the following combination of characters: (1) Embolic apophysis developed, with a semicircular membrane distally, tip of embolic apophysis significantly longer than embolic tip. (2) Ventral RTA blunt, clavate-shaped in lateral view, dorsal RTA slightly longer than ventral RTA, with sharp end. (3) Epigynal pockets running from posterior-lateral to medio-anterior. (4) Margins of lobal septum straight, almost extending to the posterior margin of epigyne, roughly forming a triangle in the median epigyne. (5) Posterior part of spermathecae reduced, significantly narrower than glandular appendage. (6) Internal ducts running parallel along median line.
**Species included.** Twelve species are included in this group: *S. anguina* Liu, Li & Jäger, 2008 (♂♀), *S. bifurca* Grall & Jäger, 2020 (♂♀), *S. bispina* Grall & Jäger, 2020 (♂), *S. fornicata* Liu, Li & Jäger, 2008 (♀), *S. improcera* Zhong et al., 2019 (♂♀), *S. lata* Zhong et al., 2019 (♀), *S. longicymbialis* Grall & Jäger, 2020 (♂♀), *S. mamillata* Zhong, Cao & Liu, 2017 (♂♀),

*S. nanphagu* Grall & Jäger, 2020 (♀), *S. phiset* Grall & Jäger, 2020 (♀), *S. rotunda* Grall & Jäger, 2020 (♀) and *S. tuber* Grall & Jäger, 2020 (♀).

**Distribution.** China (Yunnan) (Fig. 2), Brunei (Belait district), Myanmar (Southern Shan State) and Thailand (Chiang Mai Province).

### *Sinopoda globosa*-group

(Figure 4)

**Diagnosis.** This group can be recognized by the following combination of characters: (1) Subdistal embolus with a triangular projection. (2) Ventral RTA wide, blunt, dorsal RTA thinner, longer. (3) Internal ducts running parallel along median line. (4) Spermathecae with ovate posterior parts.

**Species included.** Six species are included in this group: *S. globosa* Zhang, Zhang & Zhang, 2015 (♂♀), *S. longiducta* (♂♀), *S. mi* Chen & Zhu, 2009 (♂♀), *S. ovata* Zhong et al., 2019 (♂♀), *S. triangula* Liu, Li & Jäger, 2008 (♂♀) and *S. yaanensis* (♂♀).

**Distribution.** China, Mountains around Sichuan Basin (Chongqing, Guizhou, Sichuan, Shanxi, Yunnan) (Fig. 2).

### *Sinopoda tumefacta*-group

(Figure 5)

**Diagnosis.** This group can be recognized by the following combination of characters: (1) Embolus filiform, almost straight with a reduced embolic apophysis. (2) Ventral RTA well developed and strong, dorsal RTA longer than ventral RTA. (3) Lateral lobes fused, with almost horizontal margins anteriorly. (4) Posterior parts of spermathecae swollen.

**Species included.** Six species are included in this group: *S. crassa* Liu, Li & Jäger, 2008 (♀), *S. dehiscens* Zhong et al., 2019 (♀), *S. erromena* Zhong et al., 2019 (♀), *S. tumefacta* Zhong et al., 2019 (♂♀), *S. yanlingensis* Zhong et al., 2019 (♂♀) and *S. yaojingensis* Liu, Li & Jäger, 2008 (♂♀).

**Distribution.** China (Hunan, Jiangxi, Yunnan) (Fig. 2).

### *Sinopoda campanacea* (Wang, 1990)

*Heteropoda campanacea* Wang, 1990: 7, Figs. 1–5 (Description of male and mismatched female).

*Sinopoda campanacea* Song, Zhu & Chen, 1999: 469, Figs. 269O, 270A (Description of male and mismatched female).

*Sinopoda campanacea* Jäger, 1999: 21 (Transfer from *Heteropoda*).

*Sinopoda campanacea* Yin et al., 2012: 1238, Figs. 663A–E (Description of male and mismatched female).

*Sinopoda changde* Zhong et al., 2019: 19, Figs. 13A–E, 14A–F, 15A–D (Description of male and female), **new synonym**.

**Material examined.** 5 males and 14 females (CBEE) from Hupingshan National Nature Reserve (N30.11°, E110.78°), Changde City, Hunan Province, China, 1395 m, 2017.VI.16 to 2020.VIII.1, Yang Zhong & Yang Zhu leg. 5 males and 5 females (CBEE) from Tianpingshan Scenic Area (N29.79°, E110.09°), Zhangjiajie City, Hunan Province, China, 1,503 m, 2017.VI.20, Yang Zhong & Yang Zhu leg. 4 males and 2 females (CBEE) from Taibai

Mountain (N34.06°, E107.89°), Tangyu Town, Mei County, Baoji City, Shanxi Province, China, 1,340 m, 2017.V.10, Yang Zhong & Zichang Li leg.

**Diagnosis.** *S. campanacea* is similar to *S. serpentembolus* in having strongly curved and sheet-shaped embolic apophysis, developed RTA with short and broad vRTA, longer dRTA in male, the epigyne with anterio-lateral margins of lateral lobes almost parallel with posterior margin of epigyne in female, but can be distinguished from the latter by the following characters: (1) The sperm duct of *S. campanacea* is almost straight, but significantly curved in *S. serpentembolus*. (2) The tegular apophysis is absent in *S. campanacea*, but present and located posteriorly in *S. serpentembolus*. (3) The glandular appendages are widely separated from posterior part of internal duct system in *S. campanacea*, but distinctly close with each other in *S. serpentembolus*.

**Description.** For details see *S. changde* *Zhong et al. (2019)*.

**Remarks.** *S. changde* is proposed as the new synonym of *S. campanacea* based on the following evidences: (1) *S. changde* was also collected in Tianpingshan Scenic Area where the holotype of *S. campanacea* was collected (Fig. 2). (2) The same RTA, the curved and sheet-shaped embolic apophysis, the short and slender embolic tip indicate that the male of *S. changde* belongs to *S. campanacea*. (3) *Zhong et al. (2019)* indicated that the main difference in the male palp between *S. changde* and *S. campanacea* was the palpal tegulum covering proximal part of embolus in *S. changde* but not in *S. campanacea*, we found it was due to the photos taken at different angle. (4) *Zhong et al. (2019)* proposed *S. changde* as a new species mainly based on the difference between *S. changde* and *S. campanacea* is in the female genitalia, while the matched individuals of *S. changde* (including two females and one male collected from different localities) are strongly monophyletic in the molecular phylogeny (Fig. 2). Therefore, we consider that the originally assigned female of *S. campanacea* should be another new species which is described as a new species in the following part.

### *Sinopoda papilionacea* Liu sp. nov.

urn:lsid:zoobank.org:act:FE072552-5B7A-4E32-967B-45A265B38BCA

*Heteropoda campanacea* Wang, 1990 : 7, Figs. 4–5 (Description of mismatched female).
*Sinopoda campanacea* Song, Zhu & Chen, 1999: 469, Fig. 270A (Description of mismatched female).
*Sinopoda campanacea* Jäger, 1999: 21 (Transfer from *Heteropoda*).
*Sinopoda campanacea* Yin et al., 2012: 1238, Figs. 663D–E (Description of mismatched female).

**Material (not examined, due to the loss of original type materials).** CHINA: Hunan Province: female, Tianping Mountain, Sangzhi County, Zhangjiajie City, 1984.VIII.21, Jiafu Wang & Yongjing Zhang leg. We didn't collect this species in the type locality and the type specimen described by *Wang (1990)* has been lost.

**Etymology.** The specific name is derived from the Latin adjective *papilionaceus, -a, -um*, meaning "butterfly-shaped", referring to the papilionaceous shape of internal duct systems.

**Diagnosis.** This new species can be distinguished from other *Sinopoda* species by the papilionaceous shape of internal duct systems based on the original illustrations from *Wang* (*1990*: 7, Fig. 5).

**Description (based on the illustrations and description** *Wang, 1990*). Mediaum sized Heteropodinae. PL 5.8, PW 4.8; OL 6.1, OW 4.0. Cheliceral furrow with 3 anterior and 4 posterior teeth. Dorsal prosoma deep yellowish-brown, with a yellow spot in the middle part. Fovea and redial furrows distinctly dark brown. Dorsal opisthosoma brownish black, and a yellow triangular macula in posterior part. Ventral opisthosoma yellow.

**Female genitalia:** Epigynal field wider than long, without anterior bands. Lobal septum narrow. Anterior and posterior margins of lateral lobes almost parallel. Internal duct system anteriorly touching each other at the median line but posteriorly and widely separated. Posterior part of internal duct system slightly wider than anterior part. Fertilization ducts arising posterio-laterally (*Wang, 1990*: 7, Figs. 4–5).

**Distribution.** Hunan Province, China.

**Remarks.** We didn't examine the holotype specimen of *S. papilionacea*, because the type specimens may be lost. Female of *S. papilionacea* is easily identified as a new species according to its special internal duct system based on the original illustrations.

### *Sinopoda serpentembolus* Zhang et al., 2007
(Figures 7 and 8)

*Sinopoda serpentembolus* Zhang et al., 2007: 251, Figs. 1–6 (Description of male, female may be *S. campanacea*).

*Sinopoda serpentembolus* Zhu & Zhang, 2011: 418, Figs. 298A–F (Description of male, female may be *S. campanacea*).

**Material examined.** 3 males and 5 females (CBEE) from Baotianman National Nature Reserve (N33.50°, E111.93°), Neixiang County, Nanyang City, Henan Province, China, 1300 m, 2017.VI.16, Yang Zhong & Zichang Li leg. 2 males and 4 females (CBEE) from Laojunshan Scenic Area (N33.74°, E110.63°), Luanshan County, Luoyang City, Henan Province, China, 860 m, 2017.IV.27, Yang Zhong & Zichang Li leg. 8 males (CBEE) from Taibai Mountain (N34.06°, E107.89°), Tangyu Town, Mei County, Baoji City, Shanxi Province, China, 1340 m, 2017.V.10, Yang Zhong & Zichang Li leg.

**Diagnosis.** See the above diagnosis under *S. campanacea*.

**Description. Male.** See *Zhang et al. (2007)*.

**Female (from Baotianman, China).** Mediaum sized Heteropodinae. PL 5.4, PW 4.8; AW 3.0; OL 5.7, OW 3.4. Eyes: AME 0.20, ALE 0.35, PME 0.23, PLE 0.36, AME–AME 0.24, AME–ALE 0.10, PME–PME 0.40, PME–PLE 0.54, AME–PME 0.42, ALE–PLE 0.46, CH AME 0.24, CH ALE 0.28. Spination: Palp: 131, 001, 2121, 1014; Fe: I–III 323, IV 331; Pa: I–IV 001; Ti: I–III 2026IV 2226; Mt: I–II 1014, IV–IV 3036. Measurements of palps and legs: Palps 6.6 (2.1, 0.8, 1.3, –, 2.4); I 15.2 (4.6, 1.6, 3.8, 3.9, 1.3); II 15.7 (4.8, 1.8, 4.1, 3.8, 1.2); III 13.0 (4.1, 1.4, 3.1, 3.2, 1.2); IV 14.3 (4.1, 1.7, 3.7, 3.5, 1.3). Leg formula: II-I-IV-III. Cheliceral furrow with 3 anterior and 4 posterior teeth, and with ca. 22 denticles. Dorsal prosoma deep yellowish-brown, with yellow submarginal transversal band posteriorly, fovea and redial furrows distinctly dark brown. Dorsal opisthosoma yellow-brown, covered

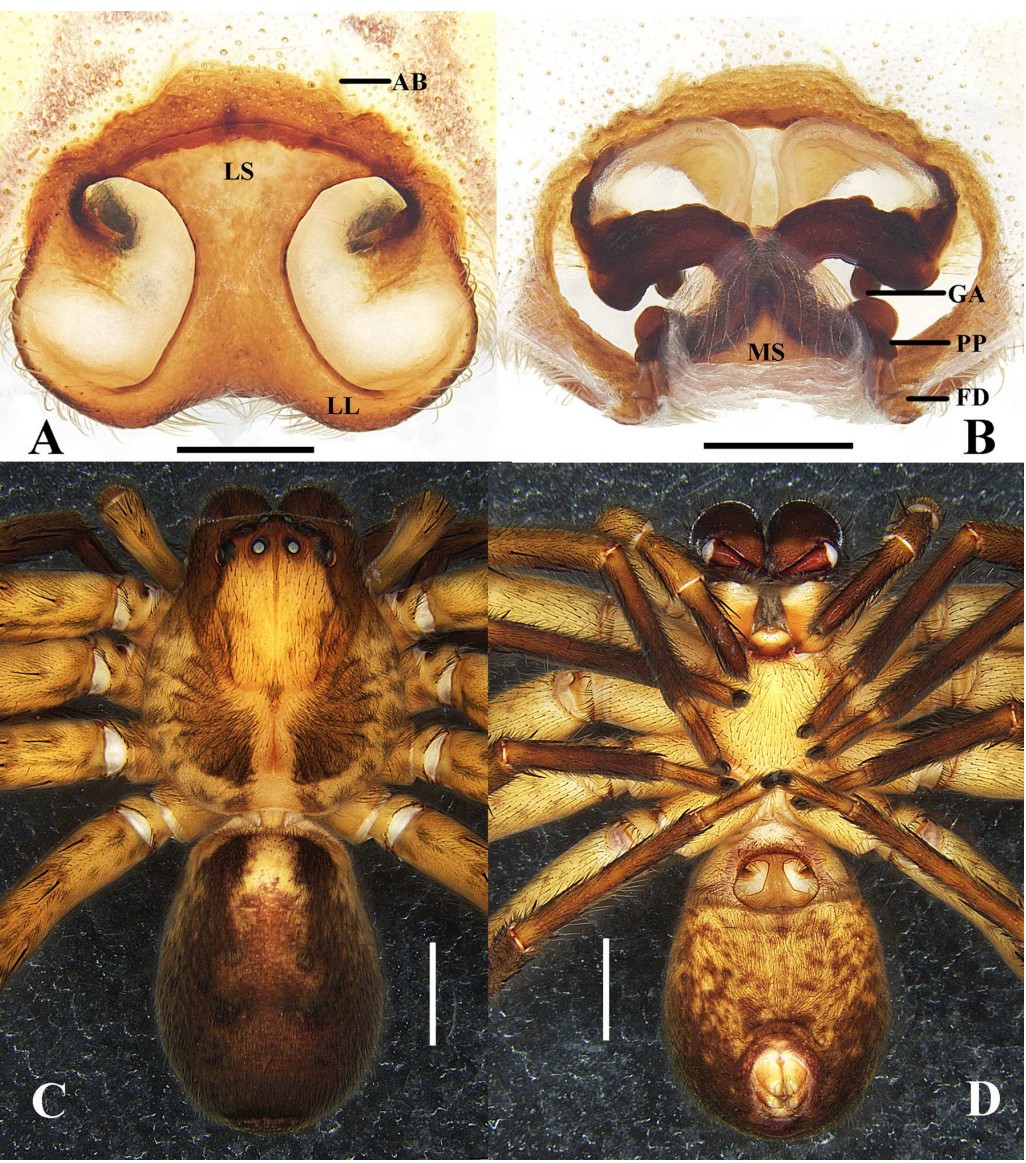

**Figure 7** *Sinopoda serpentembolus* (**Zhang et al., 2007**). (A) epigyne, ventral; (B) vulva, dorsal; (C–D), female habitus ((C) dorsal; (D), ventral). Abbreviations: AB, anterior bands; FD, fertilization duct; GA, glandular appendage; LL, lateral lobes; LS, lobal septum; MS, membranous sac; PP, posterior part of spermathecae; Scale bars: A–B 0.5 mm; C–D 2 mm.

by brown hairs. Ventral opisthosoma uniformly yellowish-brown with some irregular. Legs yellowish-brown, with dark setae.

**Female genitalia:** Epigynal field wider than long, without anterior bands and slit sensilla. Lobal septum anteriorly around 4/5 of epigyne width, anterior part wider than middle part. Lateral lobes fused. Anterior part of internal ducts system diverging. Glandular appendages long and distinctly curved. Posterior part of spermathecae, bulging laterally, fertilization ducts arising posterio-laterally (Figs. 7A–7B).

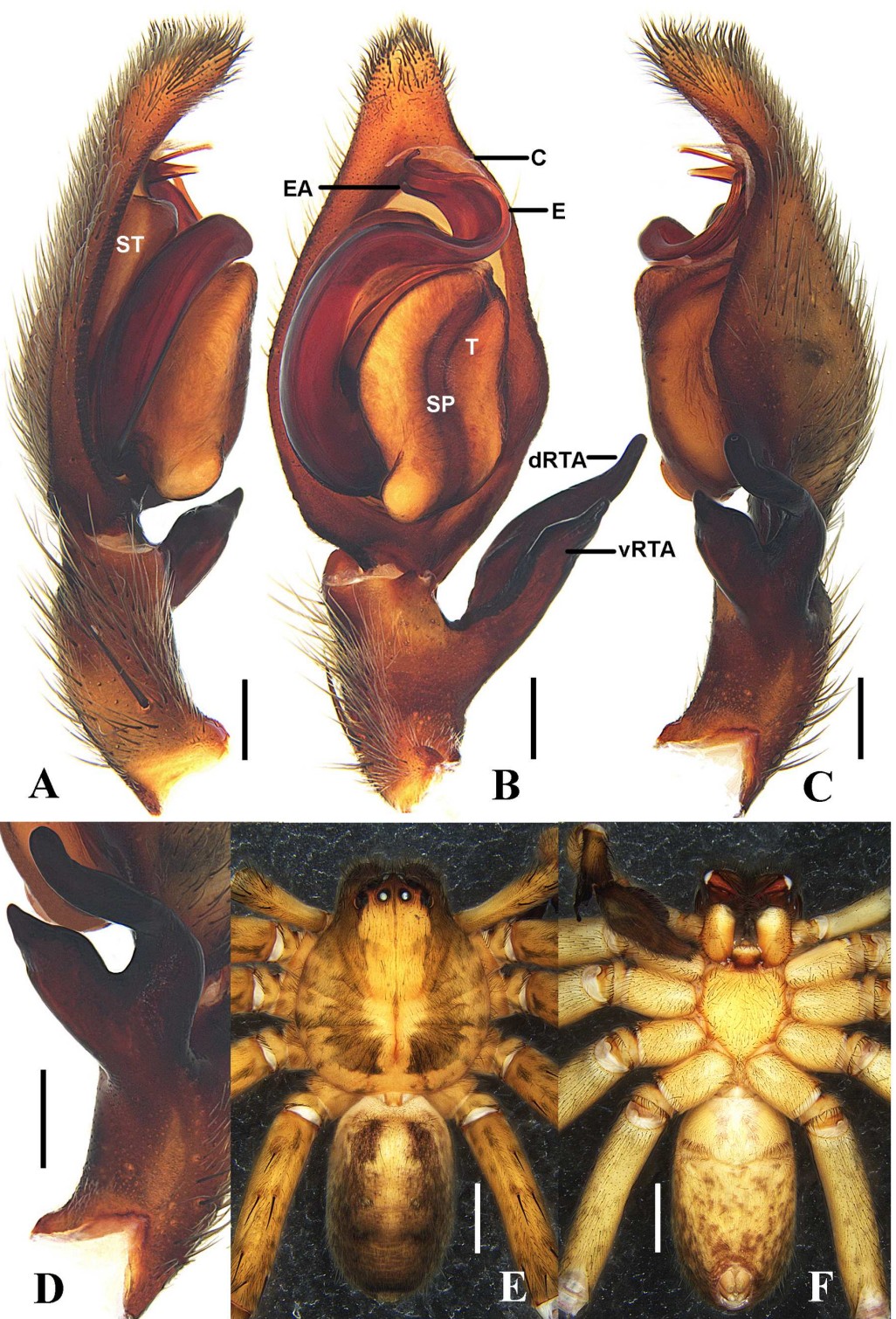

**Figure 8** *Sinopoda serpentembolus* ( Zhang et al., 2007). (A–C), left male palp (A, prolateral; B, ventral; C, retrolateral); (D), left male palpal tibia, retrolateral; (E–F), male habitus (E, dorsal; F, ventral). Abbreviations: C, conductor; dRTA, dorsal retrolateral tibial apophysis; E, embolus; EA, embolic apophysis; SP, spermophor; ST, subtegulum; T, tegulum; vRTA, ventral retrolateral tibial apophysis; Scale bars: A–D 0.5 mm; E–F 2 mm.

**Remark.** The originally matched female is very similar to *S. campanacea* in having the epigyne with anterio-lateral margins of lateral lobes almost parallel with posterior margin of epigyne, short glandular appendages, longer posterior parts of internal duct systems and having the same distribution in Taibai Mountain. However, there are also some subtle differences between them as follows: 1. The lobal septum is narrower than that of *S. campanacea*. 2. The paired internal ducts are juxtaposed medially but widely separated in *S. campanacea*. It is difficult to be sure if this is intraspecific variation or represents two species based only on morphological data. We tentatively place this specimen in *S. campanacea* here. This problem may be dealt with when we collect the fresh individuals to do the molecular analysis in the future.

**Distribution.** Henan Province and Shanxi Province, China (Fig. 2).

### *Sinopoda wuyiensis* Liu sp. nov.

urn:lsid:zoobank.org:act:E7566CCC-DE55-49A5-B9AF-0EFC5A906AB5

(Figure 9)

**Holotype.** CHINA: Fujian Province: female (CBEE) from Wuyishan National Reserve (N27.58°, E117.48°), Wuyishan City, 1300 m, 2013.VII.17, Xiaowei Cao & Yang Zhong leg.

**Paratypes.** 6 females (CBEE), same data as for holotype.

**Etymology.** 'Wuyi' refers to the type locality of this species, Wuyishan National Reserve.

**Diagnosis.** This new species can be separated from other *Sinopoda* species by the following combined characters: 1. The lateral lobes fused with each other posteriorly, with their anterior and posterior margins almost parallel. 2. Posterior part of internal duct system almost same wide as anterior part.

**Description. Male.** Unknown.

**Female (holotype).** Mediaum sized Heteropodinae. PL 5.6, PW 5.0; AW 2.7; OL 6.6, OW 3.8. Eyes: AME 0.22, ALE 0.35, PME 0.24, PLE 0.36, AME–AME 0.27, AME–ALE 0.12, PME–PME 0.37, PME–PLE 0.52, AME–PME 0.43, ALE–PLE 0.46, CH AME 0.24, CH ALE 0.26. Spination: Palp: 131, 101, 2121, 1012; Fe: I–III 323, IV 331; Pa: I–IV 101; Ti: I–II 2126, III–IV 2326; Mt: I–II 1014, IV–IV 3036. Measurements of palps and legs: Palps 7.4 (2.5, 1.1, 1.4, –, 2.4); I 18.0 (5.2, 2.0, 4.6, 4.6, 1.6); II 19.9 (5.7, 2.6, 5.6, 4.6, 1.5); III 16.3 (4.8, 2.2, 4.4, 3.4, 1.5); IV 17.6 (5.0, 2.2, 4.7, 4.2, 1.5). Leg formula: II-I-IV-III. Cheliceral furrow with 3 anterior and 4 posterior teeth, and with ca. 20 denticles. Dorsal prosoma yellowish-brown, medio-laterally with brown semicircular-pattern, posterior margins dark, with shallow fovea and redial furrows. Dorsal opisthosoma greyish-brown with three pairs of dark patches laterally. Ventral opisthosoma yellowish-brown. Legs yellowish-brown, with dark spots (Figs. 9C–9D).

**Female genitalia:** Epygnal field wider than long, with thin anterior bands. Lobal septum anteriorly around 1/8 of epigyne width. Lateral lobes fused. Internal ducts partly running parallel along the median line. Glandular appendages as wide as posterior part of spermathecae, anterior part of internal duct system narrower than posterior part. Fertilization ducts arising posterio-laterally (Figs. 9A–9B).

**Distribution.** Fujian Province, China (Fig. 2).

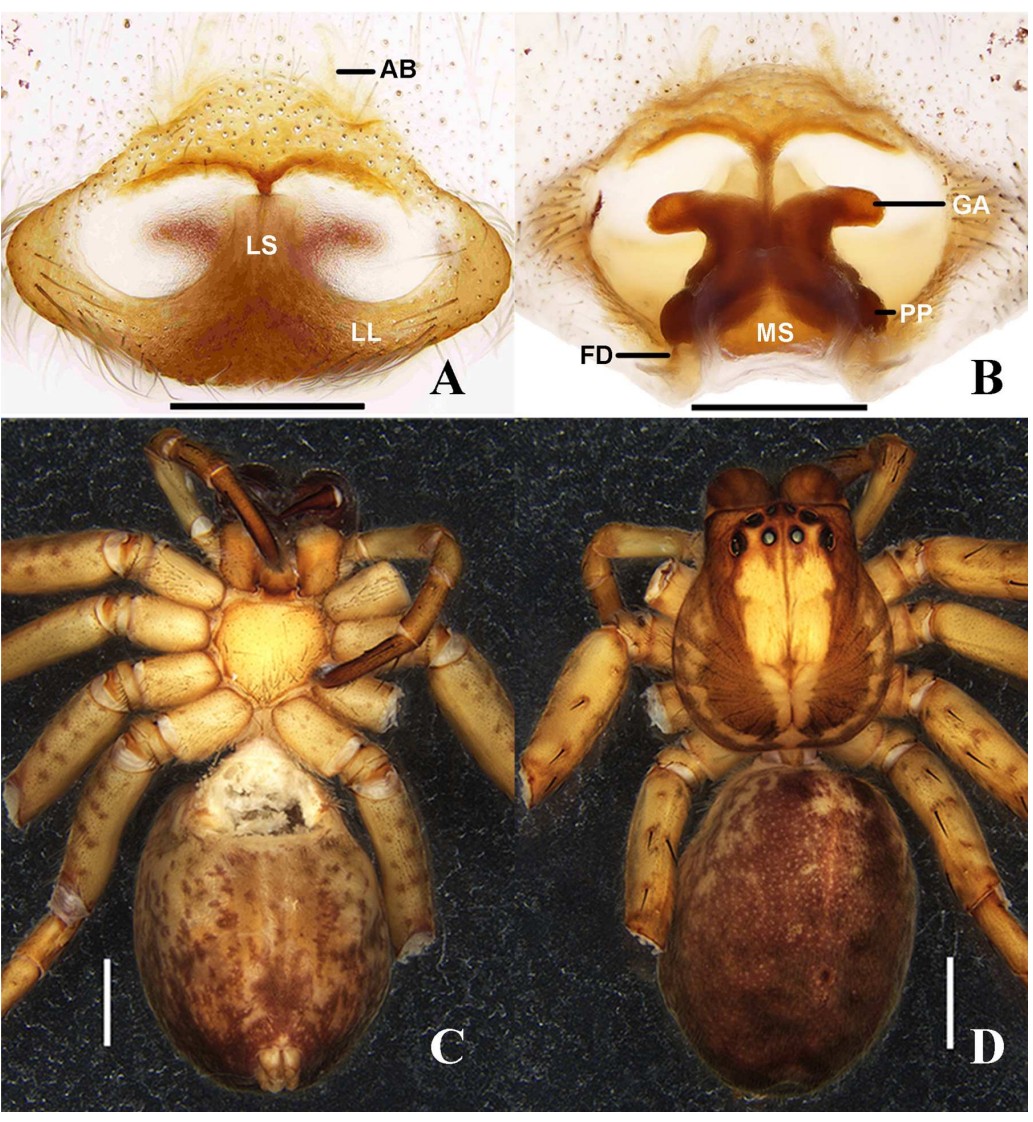

**Figure 9** *Sinopoda wuyiensis* **Liu sp. nov.** (A) epigyne, ventral; (B) vulva, dorsal; (C–D), female habitus (C, dorsal; D, ventral). Abbreviations: AB, anterior bands; FD, fertilization duct; GA, glandular appendage; LL, lateral lobes; LS, lobal septum; MS, membranous sac; PP, posterior part of spermathecae; Scale bars: A–B 0.5 mm; C–D 2 mm.

## Abbreviations

| | |
|---|---|
| **ALE** | anterior lateral eyes |
| **AME** | anterior median eyes |
| **AB** | anterior bands |
| **AW** | anterior width of prosoma |
| **C** | conductor |
| **CH** | clypeus height |
| **dRTA** | dorsal retrolateral tibial apophysis |
| **E** | embolus |

| EA | embolic apophysis |
| FD | fertilization duct |
| GA | glandular appendage |
| LL | lateral lobes |
| LS | lobal septum |
| MS | membranous sac |
| PLE | posterior lateral eyes |
| PME | posterior median eyes |
| PL | prosoma length |
| PP | posterior part of spermathecae |
| PW | prosoma width |
| ST | subtegulum |
| SP | spermophor |
| T | tegulum |
| vRTA | ventral retrolateral tibial apophysis |
| I, II, III, IV | legs I to IV |

**Collections**

| CBEE | Centre for Behavioural Ecology and Evolution, College of Life Sciences, Hubei University, Wuhan, China |

### Funding

This study was financially supported by the National Natural Sciences Foundation of China (NSFC-31572236/31970406/31772420/32000303). It was also supported by the Sciences Foundation of Hubei Province (2019CFB248) and a PhD grant from Hubei University Science and Technology (BK201811) to Yang Zhong. The funders had no role in study design, data collection and analysis, decision to publish, or preparation of the manuscript.

### Grant Disclosures

The following grant information was disclosed by the authors:
National Natural Sciences Foundation of China:
NSFC-31572236/31970406/31772420/32000303.
Sciences Foundation of Hubei Province: 2019CFB248.
Hubei University Science and Technology: BK201811.

### Competing Interests

The authors declare there are no competing interests.

## Author Contributions

- He Zhang performed the experiments, analyzed the data, prepared figures and/or tables, authored or reviewed drafts of the paper, and approved the final draft.
- Yang Zhong performed the experiments, analyzed the data, prepared figures and/or tables, and approved the final draft.
- Yang Zhu performed the experiments, prepared figures and/or tables, and approved the final draft.
- Ingi Agnarsson conceived and designed the experiments, analyzed the data, authored or reviewed drafts of the paper, and approved the final draft.
- Jie Liu conceived and designed the experiments, analyzed the data, prepared figures and/or tables, authored or reviewed drafts of the paper, and approved the final draft.

## DNA Deposition

The following information was supplied regarding the deposition of DNA sequences:

The sequence data is available in GenBank (accessions in Table S1).

## Data Availability

The raw measurements are available in the Supplemental File.

## New Species Registration

The following information was supplied regarding the registration of a newly described species:

Publication LSID: urn:lsid:zoobank.org:pub:DE32C06B-FB70-497D-A690-74DED52939DB

Sinopoda LSID:
urn:lsid:zoobank.org:act:CC11776B-A000-4C5B-B9BF-A37FE44EE03A

Sinopoda papilionacea sp. nov. LSID:
urn:lsid:zoobank.org:act:FE072552-5B7A-4E32-967B-45A265B38BCA

Sinopoda wuyiensis sp. nov. LSID:
urn:lsid:zoobank.org:act:E7566CCC-DE55-49A5-B9AF-0EFC5A906AB5.

## Supplemental Information

Supplemental information for this article can be found online at http://dx.doi.org/10.7717/peerj.11775#supplemental-information.

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
