# Peer review of "A molecular phylogeny of the Chinese Sinopoda spiders (Sparassidae, Heteropodinae): implications for taxonomy"

_PeerJ, doi:10.7717/peerj.11775_

## Round 0.1 · original submission · Major Revisions

Dear Dr. Zhang and colleagues:

Thanks for submitting your manuscript to PeerJ. I have now received three independent reviews of your work, and as you will see, the reviewers raised some minor concerns about the research. Despite this, these reviewers are very optimistic about your work and the potential impact it will have on research studying Chinese sinopods and heteropodine systematics. Thus, I encourage you to revise your manuscript, accordingly, taking into account all of the concerns raised by both reviewers.

While the concerns of the reviewers are relatively minor, this is a major revision to ensure that the original reviewers have a chance to evaluate your responses to their concerns. There are many suggestions, which I am sure will greatly improve your manuscript once addressed.

IMPORTANTLY: Avoid designating new holotypes.

Please revise your experimental design for clarity. Your methods should be clearly outlined, and your experiments should be repeatable. All datasets must be made available if the work cannot be repeatable.

Please also note that reviewers 1 and 2 have included marked-up versions of your manuscript.

Therefore, I am recommending that you revise your manuscript, accordingly, taking into account all of the issues raised by the reviewers.

I look forward to seeing your revision, and thanks again for submitting your work to PeerJ.

Good luck with your revision,

-joe

Reviewer 1 ·

Basic reporting

I indicated in the pdf file some unclear statement or too complex sentences which can be split to improve the flow of the text

Otherwise, no principal comments

Experimental design

I highly recommend to use IQtree for ML analysis, please add a comment on bootstrap parameters

You are not allowed to designate a new HOLOTYPE

Validity of the findings

The study has not clearly separated results and discussion. As a result Conclusions replace discussion (too long, partly detailed, no clear, strong summary)

Additional comments

Congratulation to valuable contribution which makes classification much stronger.
Please, consider the comments in enclosed pdf file

Annotated reviews are not available for download in order to protect the identity of reviewers who chose to remain anonymous.

·

Basic reporting

1. Just a few typos and using terms errors which are mentioned in the text directly
2. one reference missing
3. structure and figures have high quality
4. the taxa names in the Phylogenetic tree should be edited and fonts enlarged for better tracing by the reader, one suggestion is to split the tree into two pages

Experimental design

good but would be better if they add data of Sinopoda species from outside China indeed. But still useful and informative for the development of the study area

Validity of the findings

worth publishing, but consider facts below please:
1. using the term "first study" is not necessary since Sinopoda was treated in major works before
2. Your finding suggests polyphyly of okinawana-group, you may remove those species which you find out of the clade and propose new species group names for them. But you can not reject or do any kind of nomenclatural act for species group name. I suggest let the species-group name okinawana remain for those species from Japan (where the original sp-group name proposed).
more info inside the text's comments.

Additional comments

as bove

·

Basic reporting

Review for peerj-57907

In this manuscript, the authors study the spider genus Sinopoda with a focus on Chinese species. Using a Sanger-based molecular phylogeny, they propose a first molecular phylogeny for the group. This phylogeny allows the authors to revise the taxonomy and propose taxonomic changes or create new species groups.

The study has many merits as it represents a first attempt to fill in the knowledge gap of spider phylogenetics and taxonomy, especially for a complex group like this one. I have learnt a lot because I’m not a spider specialist. So I think the paper has a great potential for a large audience since it addresses the difficult topic of classifying / ranking specimens and species with morphology only. Molecular data should be helpful and is apparently helpful here. I have very little to say but I have been a bit disappointed by the low quantity of details for the phylogenetic analyses.

Experimental design

The experimental design is totally appropriate for the study. In short, the authors have extensively sampled across China and even Southeast Asia to cover a large range of the Sinopoda species. Although the sampling has many gaps, this work represents a first attempt, and will probably be complemented in a near future. They have sequenced a number of nuclear and mitochondrial markers, which are useful for reconstructing molecular phylogenies of spiders. The morphological study of the group is thorough. High-quality pictures of different anatomical parts are clear and allow diagnostic identification for the readers.

Validity of the findings

Given the taxon sampling and molecular data, the phylogeny is quite robust and well resolved. Such a phylogenetic framework can allow stressing general conclusions and making taxonomic changes and revisions. I have some comments below to improve the presentation and robustness of the phylogenetic analyses.
However, as it seems obvious, I am not a spider specialist and I have not been able to assess this part of the study. I hope the other(s) reviewer(s) have been able to evaluate the taxonomic implications of the results.

Additional comments

Major comments

(1) All the steps to prepare the phylogenetic dataset are good to me. The maximum likelihood analyses are particularly reduced to a single sentence. There is much more to do here, and I would encourage the authors to look at IQ-TREE for this type of analysis. It’s very easy and fast. It also provides interesting results with the SH test for topological hypotheses.

(2) What is the global level of support for the phylogeny of Sinopoda? I mean what is the average or median node support with posterior probabilities and bootstrap values?

(3) Why not providing a dated phylogeny? I understand the focus is on the taxonomy and a first molecular phylogeny, but it could be an additional source of information to see the divergence times. They can further help to revise the systematics.

(4) Although the work is good, I am wondering how much these conclusions can change when more species are added. You have sampled 29% of all the species known to date, but we don’t know where the missing species will be in the phylogeny. So, are they going to change your conclusions? Perhaps you should include a cautionary note toward the end of the manuscript to state that incorporating new species can alter some of the conclusions.

(5) The molecular dataset is available, which is great. However, I would recommend depositing the MrBayes nexus file instead for both better reproducibility of the analyses and better data presentation.

---

## Round 0.2 · Minor Revisions

Dear Dr. Zhang and colleagues:

Thanks for revising your manuscript. The reviewers are very satisfied with your revision (as am I). Great! However, there are a few minor edits to make. Please address these ASAP so we may move towards acceptance of your work.

Best,

-joe

Reviewer 1 ·

Basic reporting

Comments accepted. I have read again the study and made some highlights in case of formal mistakes. Italics should not be used for taxonomic authorities throughout the manuscript.

Experimental design

OK

Validity of the findings

OK

Additional comments

read very carefully for small errors again

Annotated reviews are not available for download in order to protect the identity of reviewers who chose to remain anonymous.

---

## Round 0.3 · accepted · Accept

Dear Dr. Zhang and colleagues:

Thanks for revising your manuscript based on the concerns raised by the reviewers. I now believe that your manuscript is suitable for publication. Congratulations! I look forward to seeing this work in print, and I anticipate it being an important resource for groups studying Chinese sinopods and heteropodine systematics. Thanks again for choosing PeerJ to publish such important work.

Best,

-joe

sspg